# Structures of BCL-2 in complex with venetoclax reveal the molecular basis of resistance mutations

Richard W. Birkinshaw [1,2], Jia-nan Gong[1,2], Cindy S. Luo[1,2], Daisy Lio[1,2], Christine A. White [1,2], Mary Ann Anderson[1,2,3], Piers Blombery[3,4,5], Guillaume Lessene [1,2,6], Ian J. Majewski[1,2], Rachel Thijssen[1,2], Andrew W. Roberts [1,2,3,7,8], David C.S. Huang[1,2], Peter M. Colman[1,2] & Peter E. Czabotar [1,2]

Venetoclax is a first-in-class cancer therapy that interacts with the cellular apoptotic machinery promoting apoptosis. Treatment of patients suffering chronic lymphocytic leukaemia with this BCL-2 antagonist has revealed emergence of a drug-selected BCL-2 mutation (G101V) in some patients failing therapy. To understand the molecular basis of this acquired resistance we describe the crystal structures of venetoclax bound to both BCL-2 and the G101V mutant. The pose of venetoclax in its binding site on BCL-2 reveals small but unexpected differences as compared to published structures of complexes with venetoclax analogues. The G101V mutant complex structure and mutant binding assays reveal that resistance is acquired by a knock-on effect of V101 on an adjacent residue, E152, with venetoclax binding restored by a E152A mutation. This provides a framework for considering analogues of venetoclax that might be effective in combating this mutation.

[1] Walter and Eliza Hall Institute of Medical Research, 1G Royal Parade, Parkville, VIC 3052, Australia. [2] Department of Medical Biology, The University of Melbourne, Melbourne, VIC 3052, Australia. [3] Clinical Haematology, Peter MacCallum Cancer Centre and Royal Melbourne Hospital, Melbourne, VIC 3000, Australia. [4] Department of Pathology, Peter MacCallum Cancer Centre, Melbourne, VIC 3000, Australia. [5] Sir Peter MacCallum Department of Oncology, University of Melbourne, Melbourne, VIC 3000, Australia. [6] Department of Pharmacology and Therapeutics, The University of Melbourne, Melbourne, VIC 3010, Australia. [7] Centre for Cancer Research, University of Melbourne, Melbourne, VIC 3000, Australia. [8] Victorian Comprehensive Cancer Centre, Melbourne, VIC 3000, Australia. Correspondence and requests for materials should be addressed to R.W.B. (email: birkinshaw.r@wehi.edu.au) or to P.E.C. (email: czabotar@wehi.edu.au)

The connection between aberrant cell death and cancer[1] sparked a detailed understanding of the mitochondrial pathway to apoptosis[2] and led to the first BCL-2 antagonist approved for cancer therapy, venetoclax[3,4]. BCL-2 is the founding member of the pro-survival class of proteins that includes BCL-$X_L$, BCL-W, A1/BFL-1 and MCL-1. They exert their pro-survival function by binding and restraining related members of a family of pro-apoptotic proteins—the sensors of cellular stress (the BH3-only proteins) and the effectors of apoptosis (BAX and BAK). This restraint is exerted through interactions between the helical BCL-2 Homology 3 (BH3) motifs of pro-apoptotic molecules and a cognate groove on the surface of pro-survival proteins[5]. Four or more hydrophobic amino acids on successive helical turns within the BH3 motif engage complementary pockets (P1 through P4) in this surface groove of pro-survival proteins. Blocking this interaction by targeting the binding groove with organic ligands has long held promise for cancer therapy[6,7], and venetoclax is the first realisation of that promise.

The first bona fide BH3 mimetics, ABT-737 and ABT-263 targeted multiple pro-survival family members BCL-2, BCL-$X_L$ and BCL-W, engaging with the P2 and P4 pockets in their BH3-binding groove[6,8]. Treatment with these compounds results in thrombocytopenia, an on-target toxicity due to BCL-$X_L$ inhibition, which limits their use as effective chemotherapeutics[9]. This led to a revised strategy of selectively targeting BCL-2 that resulted in ABT-199 (venetoclax)[10]. Venetoclax spares platelets but retains the ability to promote apoptosis in malignant cells dependent on BCL-2. It is among the first approved small molecule cancer therapeutics that directly blocks a protein-protein interaction[11]. Other BH3 mimetics are in development, including another BCL-2 selective compound, S55746[12].

Mutation of drug-binding sites is a common mechanism by which malignant cells evade therapies, as typified by resistance to the ABL1 tyrosine kinase inhibitors[13]. Resistance to some tyrosine kinase inhibitors can be conferred through mutation of threonine 315 to isoleucine (T315I). Rational drug design strategies have been successful in developing approved therapeutics that are effective in cases with the T315I mutation[14]. Venetoclax was approved in 2016 and as a consequence relatively few patients have been treated to date and there is a limited understanding of potential resistance mechanisms. To predict potential resistance mutants a mouse model was used to induce venetoclax tolerance in cancer cells[15]. This study identified that mutation of phenylalanine 104 (human numbering) to either leucine (F104L) or cysteine (F104C), located within BCL-2's BH3-binding groove, rendered the lymphoma cell line resistant to venetoclax. Subsequent work confirmed that these mutations can also confer resistance in models of human leukaemia and lymphoma[16], but have not yet been observed in patients. Recently a novel BCL-2 mutation was described exclusively in patients undergoing treatment with venetoclax[17]. This mutant, G101V, was found in chronic lymphocytic leukaemia (CLL) patients from the clinical trials who had initially responded to treatment but developed CLL-type clinical progression after 19–42 months[17]. Tellingly, the presence of the mutation in patient samples was predictive of clinical progression. The BCL-2 G101V mutation reduces the affinity for the drug to BCL-2 by some 180-fold. On the other hand BCL-2 G101V maintains affinity for the BH3 motif of pro-apoptotic proteins, such as BAX and BIM, and thus can still function to suppress apoptosis. By selectively reducing affinity to venetoclax the BCL-2 G101V mutation provides resistance to the therapy. To understand how the G101V and other resistance mutations can selectively decrease the affinity for venetoclax whilst retaining anti-apoptotic activity we searched for a structural rationalisation.

Currently 18 BCL-2 structures (4 NMR and 14 crystal structures) are deposited in the protein data bank (PDB)[6,10,12,18–24].

The first disclosure of venetoclax (ABT-199) in 2013 included several structures of compounds bound to BCL-2, including the structure of ABT-263 (PDB:4LVT) and an analogue of ABT-199 (PDB:4MAN)[10], but no structure of venetoclax itself with BCL-2 has yet been published.

Here we describe crystal structures of venetoclax bound to wild type BCL-2 and the BCL-2 mutants G101V and F104L. We also characterise the binding profiles of the drug to various BCL-2 mutants by surface plasmon resonance. Through these analyses we reveal the molecular mechanisms by which these mutants compromise drug-binding and, in the case of G101V, enable disease progression. These structures pave the way for rational optimisation of the venetoclax scaffold to counter this BCL-2 mutation.

## Results

**Crystal structure of BCL-2 bound to venetoclax.** Crystal structures of BCL-2 with ABT-263 and various analogues of venetoclax have been deposited in the PDB and described in the literature (Fig. 1a, b)[10]. One of those analogues is 4-[4-((4′-chloro-3-[2-(dimethylamino)ethoxy]biphenyl-2-yl)methyl)piper-azin-1-yl]−2-(1H-indol-5-yloxy)-N-((3-nitro-4-[(tetrahydro-2H-pyran-4-ylmethyl)amino]phenyl)sulfonyl)benzamide, hereafter referred to as compound 1. We obtained crystals of BCL-2 in complex with venetoclax that diffracted to high resolution (1.62 Å) in the space group $P2_12_12_1$ with one molecule in the asymmetric unit (Fig. 1a, b, Table 1). The electron density for the drug was well defined (Supplementary Fig. 1a, b) and the binding pose was in general agreement with the published structures of ABT-263 and compound 1, with the 4-chlorophenyl (CP) bound in the BCL-2 P2 pocket, the piperazine bridging the P2 and P4 pockets over residue F104 and the azaindole substitution bound in the BCL-2 P4 pocket. It was possible to model two distinct conformations with the 4–4-dimethylcyclohex-1-ene (4DM) ring flipping at the 4 and 5 positions above the BCL-2 P2 pocket and the benzamide (BA) acyl oxygen adopting two conformations above the BCL-2 P4 pocket (Fig. 1b). An interesting difference was the positioning of the venetoclax 4DM moiety, which was further from the α4 helix and more central over the P2 pocket than the equivalent rings from ABT-263 and 4MAN compounds (Fig. 1c, d, Supplementary Fig. 2, Supplementary Table 1). This small deviation in the positioning of the 4DM was unexpected as the equivalent six membered ring systems from ABT-263 and compound 1 are similar, differing only by the positioning of the gem-dimethyl group in ABT-263 (5 position) and the lack of a methyl and a phenyl ring in compound 1 (Fig. 1a).

**Structures of BCL-2 mutants bound to venetoclax.** To understand how these BCL-2 mutations compromise drug binding we solved crystal structures of both complexes (Table 1 and Fig. 2). The G101V mutation resides on the BCL-2 α2 helix packing against the α5 helix and is within the BCL-2 BH3 motif. The glycine is a conserved, defining feature of the motif. Adjacent residues A100 and D103 define boundaries of the P4 pocket so the mutation was expected to alter drug binding by changes to this region (Fig. 2a). The BCL-2 G101V:venetoclax complex crystallised in $P2_1$ spacegroup with two molecules in the asymmetric unit diffracting to 2.2 Å, with well-defined electron density for both copies of the drug (Supplementary Fig. 1). The overall structure of BCL-2 G101V was similar to WT with no significant deviations in α2–α5 core helices or α6-α8 (Fig. 2b, c). There was a minor change in orientation of the α1 helix resulting in a ~1 Å deviation at either end of the helix, but this was far from the drug binding site. The binding pose of venetoclax is conserved between WT and the G101V mutant (Fig. 2b). The P2 pocket volume was

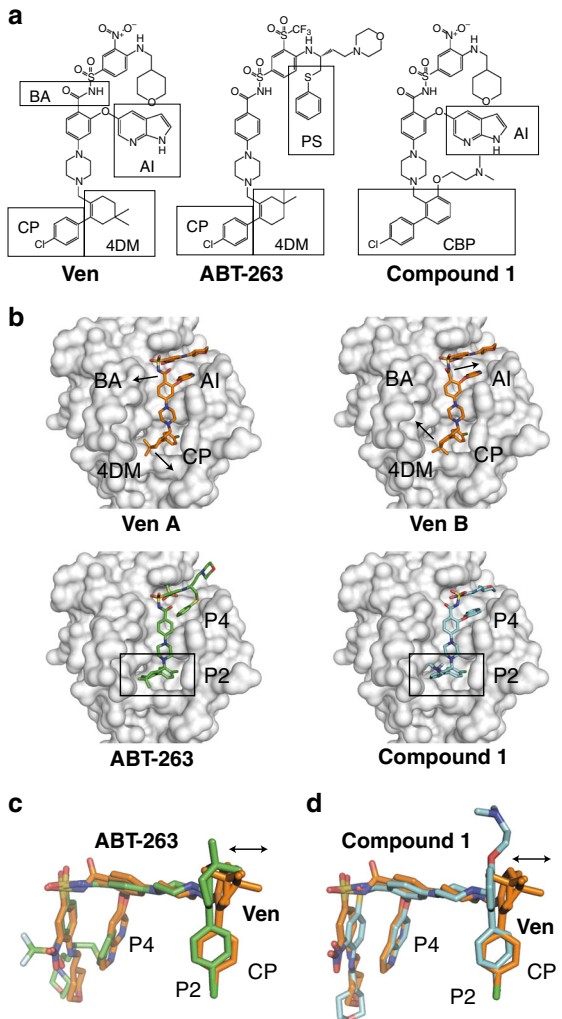

**Fig. 1** Venetoclax binding to BCL-2. **a** Chemical structures of venetoclax (Ven) and related compounds ABT-263 and compound **1**. Key features are marked, including the benzamide (BA), the 4–chlorophenyl (CP), the partially saturated 4–4-dimethylcyclohex-1-ene (4DM), the 5–5-dimethylcyclohex-1-ene (5DM), the saturated 4′-chlorobiphenyl (BP), the azaindole (AI) and phenyl sulfonyl (PS). **b** Crystal structures showing binding of the two venetoclax conformers (Ven A and Ven B, orange), and comparison to ABT-263 (green, PDB id 4LVT) and compound 1 (cyan, PDB id 4MAN) binding to the BCL-2 groove. BCL-2 is shown with surface representation (white) and P2 and P4 pockets indicated along with arrows depicting changes between venetoclax alternate conformations for BA and 4DM moieties. **c**, **d** overlays of venetoclax with ABT-263 (**c**) and compound 1 (**d**) with arrows indicating deviations compared to the venetoclax 4DM above the P2 pocket

L137 in the α4 helix (Fig. 2b, Supplementary Fig. 2, Supplementary Table 1), i.e. more similar to the BCL-2 WT complexes with ABT-263 and compound **1**. Additionally, we obtained a structure of BCL-2 G101A bound to venetoclax (Table 1), representing a milder introduction of bulk at the G101 position than the valine substitution. The BCL-2 G101A:venetoclax and BCL-2 WT:venetoclax crystals were isomorphous and there were no significant deviations in venetoclax positioning or E152 (Supplementary Fig. 2, 3, Supplementary Table 1). Despite the closer proximity of the azaindole moiety to the mutation site, its orientation in the P4 pocket was conserved between BCL-2 WT and G101V structures. Therefore, the G101V mutation appears to modulate venetoclax affinity more through its interactions with the P2 pocket than the P4 pocket.

The crystals of venetoclax complexed with BCL-2 F104L and BCL-2 WT are isomorphous (Table 1). Well-defined electron density for the drug in the mutant complex structure (Supplementary Fig. 1) suggests two conformations for the 4DM and acyl group of the BA moiety as in WT (Fig. 2d). The side chain of F104 separates the P2 and P4 pockets of BCL-2. The P4 pocket volume was maintained between WT and F104L structures (P4 pocket volume 380 Å³ for both WT and F104L). In contrast the volume of the P2 pocket increased with the F104L mutation as the leucine sidechain occupies a smaller volume than phenylalanine (Fig. 2d–f). In this structure two conformers for F112 on α3 are evident, one like WT and a second occupying some of the volume vacated by the F104L mutation. (Fig. 2c, e and f). The new F112 conformation extends into the P2 pocket, packing against L104. This inserted F112 conformation compensates for the loss of P2 pocket volume in the BCL-2 F104L mutant and is comparable to the BCL-2 WT P2 pocket volume—P2 pocket volumes of 480 Å³ (BCL-2 WT), 475 Å³ (F104L inserted conformation) and 596 Å³ (F104L conserved conformation). The occupancy of F112 refined to 0.48 for the conserved conformation and 0.52 for inserted conformation, indicating that the compensation in P2 pocket volume only occurs 50% of the time and the vacated space is unfavoured. The consequence of the F104L mutation is to alter the packing environment of the chlorophenyl moiety of the drug.

**Binding of BH3 peptides and venetoclax to BCL-2 mutants.** SPR experiments were performed using a BIMBH3 or BAXBH3 immobilised sensor surface with BCL-2 mutants as the analyte and determining venetoclax affinity by competition experiments[17,25], (Fig. 3, Table 2 and Supplementary Fig. 4, 5). We have previously reported BIMBH3 and BAXBH3 affinities for WT, G101V and F104L[17], and these were comparable to F104C, with less than 10-fold change relative to WT (Table 2). In contrast, the affinities for venetoclax differed by 25 to ~1500-fold with $K_I$ values 0.018, 3.2, 0.46 and 25 nM for WT, G101V, F104L and F104C, respectively (Table 2). This indicates that the BCL-2 mutants maintain tight binding to BH3 domains, allowing their overexpression to prevent apoptosis, whilst selectively reducing the affinity for the drug and thus providing resistance to therapy.

**The role of E152 in venetoclax affinity.** E152 moved into the base of the P2 pocket in the BCL-2 G101V:venetoclax structure (Fig. 2b, c). To test the role of E152 in reducing affinity we generated a BCL-2 G101V/E152A double mutant. Alanine does not have a Cγ or Cδ to impact the base of the P2 pocket and would allow the chlorophenyl to insert unimpeded into the P2 pocket in the G101V mutant. We repeated SPR experiments with the BCL-2 G101V/E152A double mutant and a BCL-2 E152A single mutant (Table 2 and Fig. 3c, d). The E152A single mutant had comparable binding to WT and when combined with G101V

maintained at 478 Å³ (480 Å³ for WT) as was the volume of the P4 pocket at 379 Å³ (380 Å³ for WT). Interestingly, the G101V mutation did not alter the positioning of the α2 helix relative to α5 or impact the residues defining the P4 pocket. Instead the additional bulk of the valine sidechain was accommodated by a deviation of the sidechains of Y18 on α1 and E152 on α5 (Fig. 2c). In the BCL-2 G101V:venetoclax structure E152 had a 60° rotamer change relative to WT (mm-40 for WT to tp10 for G101V), placing the sidechain Cγ in van der Waal's contact with the chlorine atom of the venetoclax chlorophenyl moiety. This conformational change in E152 in the BCL-2 G101V structure causes a small repositioning of the venetoclax 4DM and chlorophenyl moieties in the P2 pocket, moving on average 0.25 Å closer to

**Table 1 Crystal structure data collection and refinement statistics**

| | BCL-2 WT: Ven | BCL-2 G101V: Ven | BCL-2 G101A: Ven | BCL-2 F104L: Ven | BCL-2 G101V: S55746 |
|---|---|---|---|---|---|
| PDB id | 6O0K | 6O0L | 6O0P | 6O0M | 6O0O |
| Wavelength (Å) | 0.9537 | 0.9537 | 0.9537 | 0.9537 | 0.9537 |
| Resolution range (Å) | 42.4–1.62 (1.68–1.62) | 47.51–2.2 (2.28–2.20) | 43.24–1.8 (1.86–1.80) | 43.9–1.75 (1.81–1.75) | 34.29–1.998 (2.07–2.00) |
| Space group | $P2_12_12_1$ | $P2_1$ | $P2_12_12_1$ | $P2_12_12_1$ | $P2_{1benioff}$ |
| Unit cell (Å, °) | 33.73 48.51 87.32 90 90 90 | 33.15 82.01 47.51 90 90.08 90 | 33.40 48.95 86.47 90 90 90 | 33.69 48.27 87.80 90 90 90 | 37.68 68.58 64.49 90 96.67 90 |
| Total reflections | 128406 (8348) | 49434 (4673) | 82474 (5028) | 87816 (6285) | 148201 (14601) |
| Unique reflections | 18474 (1508) | 12909 (1270) | 13559 (1200) | 15022 (1427) | 22155 (468) |
| Multiplicity | 7.0 (5.5) | 3.8 (3.7) | 6.1 (4.2) | 5.8 (4.4) | 6.7 (6.6) |
| Completeness (%) | 97.77 (80.73) | 99.84 (99.53) | 98.45 (86.85) | 99.72 (97.25) | 74.31 (21.24) |
| Mean I/sigma(I) | 12.57 (2.16) | 7.60 (1.56) | 9.56 (0.82) | 10.58 (1.58) | 7.29 (0.90) |
| Wilson B-factor | 15.5 | 36.24 | 30.55 | 21.39 | 33.19 |
| R-meas | 0.1054 (0.9667) | 0.1432 (0.9676) | 0.1129 (2.024) | 0.1154 (1.119) | 0.148 (2.474) |
| R-pim | 0.03963 (0.3948) | 0.07165 (0.4947) | 0.04476 (0.9436) | 0.04728 (0.5198) | 0.05719 (0.9544) |
| $CC_{0.5}$ | 0.998 (0.604) | 0.991 (0.646) | 0.997 (0.292) | 0.996 (0.436) | 0.997 (0.596) |
| Reflections used in refinement | 18462 (1500) | 12924 (1266) | 13525 (1176) | 15011 (1417) | 16499 (468) |
| Reflections used for R-free | 905 (69) | 628 (60) | 672 (56) | 743 (69) | 816 (19) |
| R-work | 0.1633 (0.2301) | 0.1952 (0.2530) | 0.1929 (0.3681) | 0.1732 (0.2908) | 0.2167 (0.2937) |
| R-free | 0.2013 (0.2848) | 0.2278 (0.2963) | 0.2314 (0.4223) | 0.2181 (0.3273) | 0.2668 (0.3437) |
| Number of non-hydrogen atoms | 1512 | 2502 | 1313 | 1511 | 2409 |
| Macromolecules | 1264 | 2328 | 1195 | 1275 | 2288 |
| Ligands | 150 | 152 | 82 | 150 | 106 |
| solvent | 98 | 22 | 36 | 86 | 15 |
| Protein residues | 141 | 279 | 141 | 141 | 273 |
| RMS(bonds) | 0.018 | 0.003 | 0.003 | 0.004 | 0.005 |
| RMS(angles) | 1.41 | 0.59 | 0.57 | 0.72 | 0.91 |
| Ramachandran favored (%) | 99.27 | 99.26 | 100 | 100 | 98.49 |
| Ramachandran outliers (%) | 0 | 0 | 0 | 0 | 0 |
| Rotamer outliers (%) | 0.75 | 0.41 | 0 | 0.75 | 1.26 |
| Clashscore | 5.67 | 8.05 | 4.94 | 7.1 | 4.39 |
| Average B-factor (Å²) | 22.23 | 50.02 | 39.44 | 25.35 | 50.2 |
| Macromolecules | 21.2 | 50.34 | 39.44 | 24.28 | 49.9 |
| Ligands | 22.4 | 46.14 | 38.12 | 28.02 | 58.21 |

Statistics for the highest-resolution shell are shown in parenthesis

as a double mutation restored high affinity venetoclax binding, with WT binding at 18 pM, BCL-2 E152A at 27 pM and BCL-2 G101V E152A at 2 pM (Table 2 and Fig. 3c, d). The BCL-2 G101V/E152A affinity was 10-fold higher than WT, however competition SPR experiments become less accurate as the ligand $K_I$ becomes significantly tighter or weaker than the $K_D$ for the competing BimBH3 peptide; as such it is unclear whether this increase in affinity is significant. Furthermore, the E152 conformation in a BCL-2 G101A:venetoclax structure matched the WT conformation, not the G101V. The G101A mutant bound to venetoclax with a $K_I$ of 110 pM comparable to WT but distinct from G101V (Table 2 and Supplementary Fig. 4). This indicates that E152A mutation rescues high affinity for venetoclax when combined with G101V and confirms the importance of the E152 rotamer change observed in the G101V mutant. Note also that the affinity of BCL-2 G101V/E152A and BCL-2 E152A for BIMBH3 and BAXBH3 was largely unaltered compared to WT (Table 2 and Supplementary Fig. 3).

**BCL-2 G101V binding to S55746.** S55746 is another BCL-2 selective antagonist that has progressed to the clinic. The recently disclosed crystal structure of BCL-2 WT bound to S55746 revealed binding to the P1, P2 and P3 pockets[12], in contrast to venetoclax that binds principally to the P2 and P4 pockets

(Fig. 4). We tested the binding of S55746 to both BCL-2 WT and G101V by competition SPR (Table 2, Fig. 3 and Supplementary Fig. 6). S55746 bound to BCL-2 WT with a $K_I$ of 0.36 nM and G101V with a 100-fold lower $K_I$ of 36 nM, in both cases > 10-fold weaker than venetoclax, $K_I$ of 0.018 nM and 3.2 nM for WT and G101V, respectively. This was confirmed in cellular assays using the B-lineage cell line KMS-12-PE (Fig. 4a). BCL-2 WT and G101V were overexpressed in the KMS-12-PE cells and S55746 LC50 concentrations were determined as $0.32 \pm 0.15\,\mu M$ for WT increasing eight-fold to $2.7 \pm 0.43\,\mu M$ with the G101V mutation.

To further investigate this we solved the structure of BCL-2 G101V bound to S55746 (Table 1). We obtained diffraction to 2.0 Å in a $P\,2_1$ spacegroup with two molecules in the asymmetric unit. The BCL-2 G101V:S55746 structure was in general agreement with the published structure of the BCL-2 WT: S55746 complex PDB ID 6GL8 (Fig. 4b, c)[12]. The P2 pocket is key for interactions with BH3 domains[26,27], which typically display a leucine residue engaging this pocket. S55746 inserts a 4-hydroxyphenyl moiety into the P2 pocket similar to the chlorophenyl of venetoclax. In that case the chlorophenyl inserts deeply into the P2 pocket causing F112 to change rotamer (venetoclax rotamer t80, S55746 rotamer m-85) and exposing E152 at the base of the P2 pocket (Fig. 4d). In the S55746 structure F112 seals the P2 pocket shielding the S55746 hydroxyphenyl from E152 (Fig. 4d). In the BCL-2 WT:S55746 structure E152 is

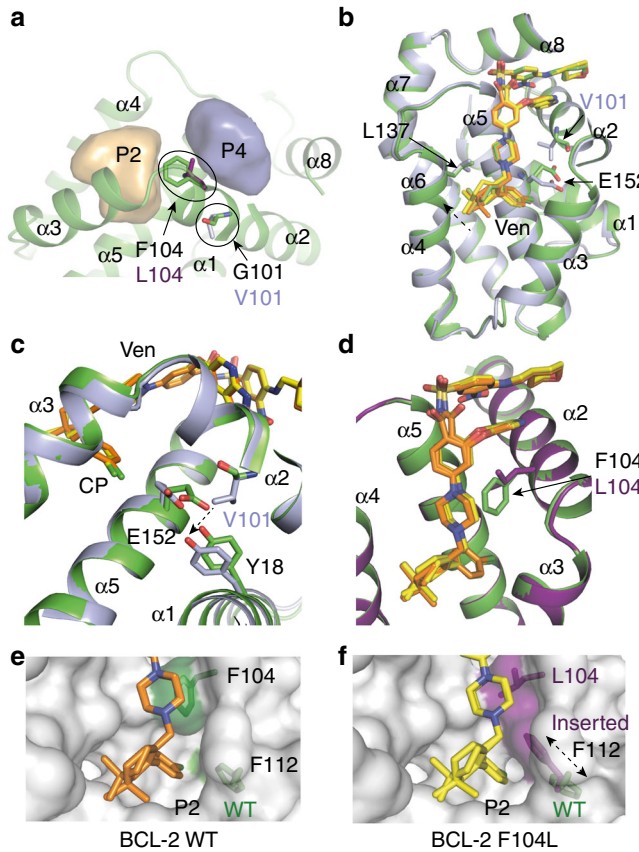

**Fig. 2** BCL-2 mutations and venetoclax binding. **a** location of G101V and F104L mutants relative to the BCL-2 groove, α1–8 helices and P2 (orange) and P4 (blue) pockets. Pockets volumes were defined by venetoclax (Ven) chlorophenyl (CP) for P2 pocket and azaindole ring for P4 pocket, see methods for further details. **b**, **c** Overlay of BCL-2 WT and BCL-2 G101V bound to venetoclax. **d** Overlay of BCL-2 WT and BCL-2 F104L bound to venetoclax. **e** Surface representation of the BCL-2 WT:venetoclax P2 pocket. **f** Surface representation of the BCL-2 F104L:venetoclax P2 pocket. In **a–f** Venetoclax is shown in orange for BCL-2 WT and yellow in mutants, BCL-2 WT in green, BCL-2 G101V in pale blue, F104L in purple and surfaces in white. Arrows indicate key residues, dashed arrows indicate residue movements or movement of venetoclax

in the tp10 rotamer configuration similar to the BCL-2 G101V: ABT-199 structure. However, in the BCL-2 G101V: S55746 structure E152 is in an unconventional rotamer, which shows most similarity to tp10, with the Cγ deviating by 40° from the conventional tp10 rotamer (Fig. 4d). We tested binding of the BCL-2 G101V/E152A double mutant to S55746 giving a $K_I$ of 5.3 nM. This was over 10-fold higher than WT ($K_I$ of 0.32 nM), but 10-fold lower than G101V ($K_I$ of 36 nM), indicating partial but not complete rescue of S55746 affinity with the double mutant (Fig. 3g). Thus, one cause of the reduction in affinity of S55746 for BCL-2 G101V can be traced to the knock-on effect of the V101 side chain against E152.

## Discussion

Currently venetoclax is approved for treatment of patients with previously treated chronic lymphocytic leukemia[4]. Venetoclax selectively inhibits BCL-2, thereby promoting apoptosis in cells refractory to conventional apoptotic cues. No crystal structure for BCL-2 binding to venetoclax has been described, although analogues have been published[10]. We have now crystallised and determined the structure of venetoclax bound to BCL-2 at high

resolution. The structure reveals subtle differences between the binding of the drug relative to the published analogues (Fig. 1), which would not have been predicted based on the previous structures. Notably, the orientation of the moiety inserted into the P2 pocket is subtly different in venetoclax compared to ABT-263 and compound **1** structures. The P2 pocket is an important determinant in selectivity of BH3 peptides and BH3 mimetics[26,28], and here it emerges as the critical feature conferring resistance to a drug-selected BCL-2 mutant, G101V.

The BCL-2 G101V mutation was exclusively identified in CLL patients with disease progression on venetoclax clinical trials, but only after many months of continuous treatment. Early detection of BCL-2 G101V by highly sensitive ddPCR assays predicted subsequent clinical disease progression. The mutation reduces the affinity of the drug for BCL-2 by ~180-fold[17]. In contrast, it only moderately reduces the affinity for BH3 peptides, allowing the mutant to still function normally as a pro-survival protein. Interestingly, G101 is part of the BCL-2 BH3 sequence. The BH3 motifs of BH3-only proteins or of BAX and BAK are ligands for a binding groove on pro-survival BCL-2 proteins, but the role of the BH3 motif in pro-survival proteins is unclear. Notably, the motif consensus sequence of ϕ1-x-x-x-ϕ2-x-x-ϕ3-G-D-x-ϕ4, where ϕ1–4 are hydrophobic amino acids, includes the largely conserved G101 of BCL-2. There are examples of BCL-2 family proteins that have either alanine or serine instead of glycine at this position, but not valine. In the multi-BH domain BCL-2 proteins the BH3 motif is in the α2 helix, with the small glycine packing against the BH1 domain in the α5 helix. The G101V mutation is adjacent to but not directly part of the P4 pocket that engages a leucine residue (ϕ4) when the BAX BH3 motif binds (Fig. 5). We could not obtain crystals for BCL-2 G101V bound to BAXBH3 peptide so cannot comment on how the mutation affects BAXBH3 interactions in this region. The azaindole ring of venetoclax, one of the key features of its selectivity for BCL-2, occupies this pocket. However, there were no differences in the P4 pocket with venetoclax bound to BCL-2 or BCL-2 G101V. Instead the additional bulk from the valine sidechain is accommodated by movements in the positions of Y18 (α1) and E152 (α5), with a distinct rotamer change for the glutamate. The E152 rotamer change in the BCL-2 G101V:venetoclax complex places the Cγ sidechain atom in contact with the chlorophenyl introducing a subtle change in the orientation of that moiety in the P2 pocket. To illustrate the connection between these small structural movements and the ~180-fold reduction in affinity for the G101V mutant, we introduced the E152A mutation on the G101V mutant and restored near-wildtype affinity for venetoclax. Additionally, introduction of an alanine at the G101 position did not provide sufficient bulk to displace E152 in the crystal structure and affinity for venetoclax was maintained for the G101A mutant. Thus the functional consequences of the G101V mutant are felt in the P2 pocket via the 'knock-on' effect of E152 repositioning forced by the additional bulk of the valine sidechain.

The BCL-2 F104L and F104C mutations were observed as venetoclax-resistance mutations in a mouse tumour model[15] and both induce drug tolerance in human cell lines[16]. Interestingly, in the initial study, human cell lines did not acquire resistance through mutation to BCL-2, instead truncation of BAX occurred preventing translocation to the mitochondria[15]. The BCL-2 F104L mutation has been observed in non-Hodgkin lymphoma[29] suggesting the mutation is viable in lymphomas, however these patients were not treated with drug and to date neither the F104L or F104C mutations have been reported in patients receiving venetoclax therapy. Here we have shown through competition SPR experiments that neither the BCL-2 F104L nor F104C mutants suffer significant reduction in binding to the BH3 peptides of pro-apoptotic BAX or BIM. In contrast, venetoclax

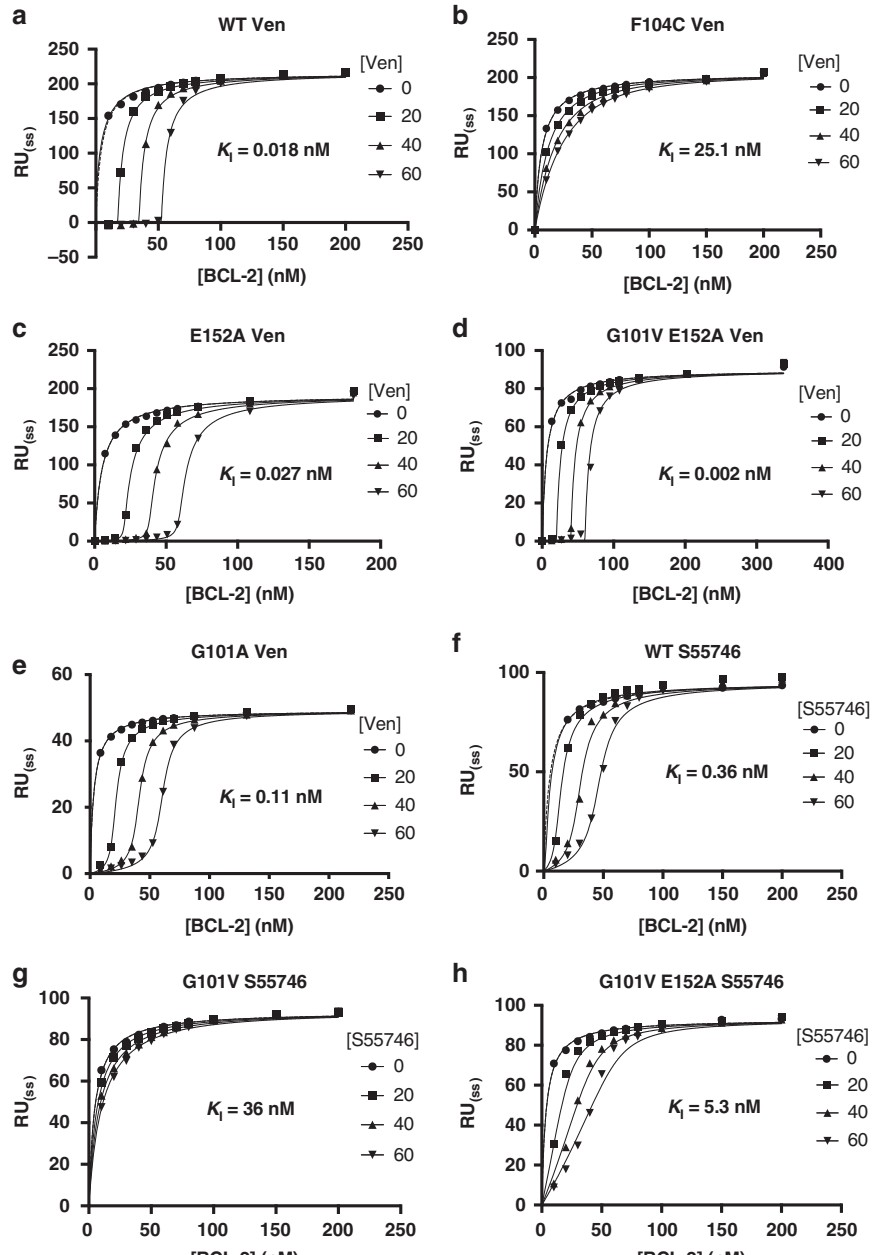

**Fig. 3** Steady-state competition SPR with BCL-2 mutants, BIMBH3 and venetoclax or S55746. SPR chip surfaces were immobilised with BIMBH3 peptide and analytes with the indicated BCL-2 mutants pre-combined with either **a–e** venetoclax (Ven) or **f–h** S55746 at the indicated concentrations (circles 0 nM, squares 20 nM, triangles point up 40 nM, triangles point down 60 nM). Response at steady-state is plotted against BCL-2 concentration, with fits from a steady-state competition SPR model shown (see methods and Supplementary Figs. 5, 6 for further details), that were used to derive the indicated mean $K_I$ values for venetoclax or S55746 binding to BCL-2. Data are representative of at least $n = 2$ independent experiments

binding decreased by ~10–500-fold (for F104L and F104C, respectively) relative to WT BCL-2. The structure of BCL-2-F104L:venetoclax presented here reveals that the P2 pocket increases in volume with the F104L mutation relative to WT (480 $\text{Å}^3$ and 596 $\text{Å}^3$ for WT and F104L, respectively). This increase in pocket volume decreases the surface complementarity between the drug and its target, likely contributing to the decrease in affinity. Binding to the F104C mutant is weaker still, and this may have thwarted attempts to obtain a crystal structure of the BCL-2 F104C:venetoclax complex despite extensive efforts. This further decrease in affinity is likely due to an even larger P2 pocket volume, as cysteine occupies a smaller volume than leucine, though other structural features may also come into play.

Pro-survival proteins prevent apoptosis by binding and sequestering the pro-apoptotic proteins. Venetoclax competes with pro-apoptotic BH3 motifs for BCL-2 binding, releasing pro-apoptotic proteins and allowing apoptosis in cells primed for death[6,8]. It is therefore required that drug-resistant mutants of BCL-2 retain the ability to bind pro-apoptotic BH3 motifs to maintain the tumour's viability. The G101V, F104L and F104C BCL-2 mutants all have this property. The critical difference between venetoclax and a BH3 helical peptide is the greater penetration of the drug into the P2 pocket, with the G101V mutant this compromises drug-binding but not BH3-binding. This is also a feature of ABT-737, one of the earliest precursors of venetoclax[26]. Analogues of venetoclax that may retain binding to

**Table 2 SPR affinity values for BH3 peptides and BCL-2 selective compounds binding to BCL-2 mutants**

| BCL2 protein | BIMBH3 $K_D$ (nM ± SD, $n \geq 2$) | BAXBH3 $K_D$ (nM ± SD, $n \geq 2$) | Ven $K_I$ (nM ± SD, $n=3$) | S55746 $K_I$ (nM ± SD, $n \geq 2$) |
|---|---|---|---|---|
| WT[a] | 0.29 ± 0.17 | 1.4 ± 0.2 | 0.018 ± 0.014 | 0.36 ± 0.02 |
| G101V[a] | 0.84 ± 0.04 | 12 ± 2 | 3.2 ± 1.1 | 36 ± 12 |
| G101A | 0.23 ± 0.04 | ND | 0.11 ± 0.03 | ND |
| F104C | 0.17 ± 0.10 | 4.0 ± 0.2 | 25.1 ± 6.3 | ND |
| F104L[a] | 0.039 ± 0.01 | 6.1 ± 1.0 | 0.46 ± 0.12 | ND |
| E152A | 0.36 ± 0.10 | 3.6 ± 0.9 | 0.027 ± 0.008 | ND |
| G101V/E152A double mutant | 0.63 ± 0.39 | 5.5 ± 0.3 | 0.0020 ± 0.0027 | 5.3 ± 4.1 |

Affinity for BIMBH3 and BAXBH3 peptides were determined by direct binding using kinetic fitting (see Supplementary Table 2 for detailed binding fit parameters). Venetoclax and S55746 by steady-state competition against a BIMBH3 peptide SPR surface with the indicated BCL-2 mutant. Mean values from at least two independent experiments with one standard deviation indicated
ND not determined
[a]previously reported[17]

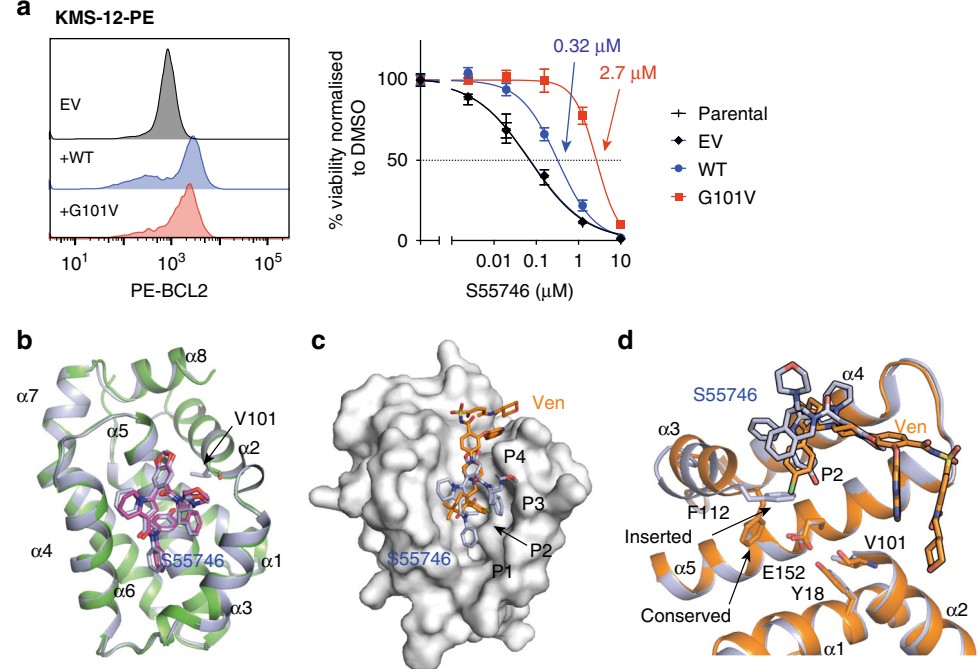

**Fig. 4** BCL-2 G101V binding to S55746. **a** Expression of BCL-2 G101V mutant reduces S55746 sensitivity in the KMS-12-PE B-lineage cell line. WT and G101V mutant BCL-2 proteins were expressed at similar levels in KMS-12-PE cells as demonstrated by the FACS profiles. In vitro sensitivity to S55746 (0–10 μM) was measured after 24 h by a CellTiter-Glo assay. Data are representative of n = 3 independent experiments with mean LC50 values ± SD (1 Standard Deviation) for empty vehicle (EV, black), WT (blue) and G101V (red) indicated. **b** Overlay of the published BCL-2:S55746 structure (green and magenta, PDB id 6GL8) with the BCL-2 G101V:S55746 structure (light blue) showing consistency in S55746 orientation. **c, d** Overlay of BCL-2 G101V structures bound to S55746 (light blue) and venetoclax (Ven, orange). **c** BCL-2 protein surface displayed from the BCL-2 G101V:S55746 structure showing P1–4 pockets. **d** Cartoon representation with key residues indicated in stick representation, inserted conformation of F112 in the BCL-2 G101V—S55746 and WT conformation from BCL-2 G101V:venetoclax structure are indicated

BCL-2 G101V include those that lack the chlorine of the chlorophenyl moiety or have it replaced with a smaller atom.

S55746 binds BCL-2 via the P1, P2 and P3 pockets and our SPR data show it binds to BCL-2 WT with 10-fold lower affinity than venetoclax. S55746 inserts a 4-hydroxyphenyl moiety into the P2 pocket but does not insert as deep into the pocket as the corresponding chlorophenyl from venetoclax. This shallow insertion allows the BCL-2 F112 on the α3 helix to insert into the BCL-2 groove extending the P3 pocket. We show here that S55746 also loses potency (~100-fold) against the BCL-2 G101V mutant and we sought a structural explanation for this. The orientation of S55746 in the BCL-2 G101V:S55746 complex structure is conserved relative to the WT structure. In both BCL-2

WT and BCL-2 G101V structures compared with S55746 E152 is shielded from the base of the P2 pocket by the insertion of F112 into the groove. Consistent with this the combination of the G101V mutation with the E152A mutation did not fully restore WT binding to S55746, in contrast to venetoclax, suggesting additional structural features underlying affinity in this case. Those features may include changes in the structure and dynamics of BCL-2 G101V prior to engaging drug, or the distribution of conformers in the ensemble of BCL-2 G101V:drug structures in solution compared to WT that are not readily detectable crystallographically[30].

Venetoclax is the first FDA-approved drug that directly targets the mitochondrial apoptotic pathway. We have solved the crystal

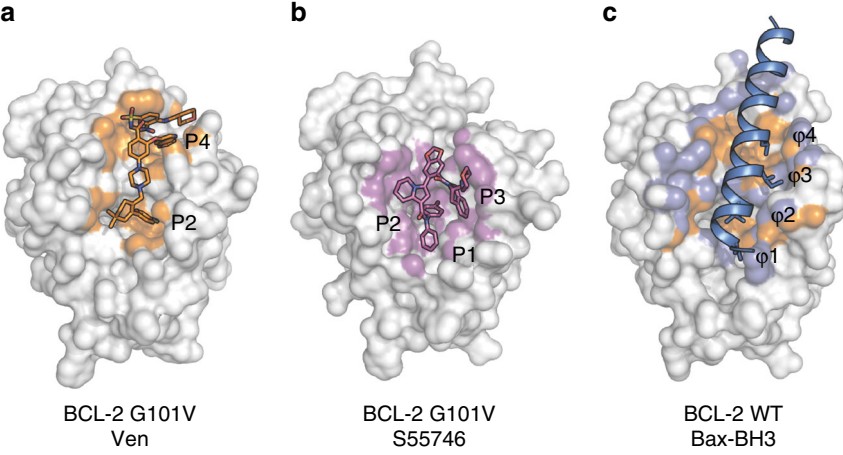

**Fig. 5** BCL-2 surface contacts with either venetoclax, S55746 or BaxBH3 peptide. Structure of BCL-2 G101V with **a** Venetoclax (Ven) or **b** S55746, or **c** BCL-2 WT with a BaxBH3 peptide. BCL-2 surface contacts are coloured according to interactions with venetoclax (orange), S55746 (magenta) or BaxBH3 peptide (slate). Key BCL-2 binding pockets (P1 through P4) indicated for each compound and hydrophobic (φ) 1–4 residues (L, L, I, L) from the BaxBH3 peptide. **c** BCL-2 surface contacts with venetoclax are coloured in orange and with BaxBH3 peptide in slate highlighting the increased surface contact area of BaxBH3 relative to venetoclax (PDB id 2XAO)

structure of the drug bound to its target, BCL-2, revealing a pose subtly different to that observed with structural analogues. We also describe the molecular basis for resistance to venetoclax observed in patients on treatment displaying the BCL-2 mutant G101V. These structures provide a basis to further optimise the venetoclax scaffold, for binding both to BCL-2 WT and the G101V mutant.

## Methods

**Compounds, peptides and primers**. All peptides were custom synthesized by Mimotopes Pty Ltd (Australia). S55746 was purchased from ProbeChem (Cat#PC-63502). Venetoclax was purchased from ActiveBiochem (#A-1231). All other chemicals, unless specified, were obtained from Sigma-Aldrich (Australia). Primer sequences are detailed in Supplementary Table 3.

**Protein expression and purification**. BCL-2 WT construct, expression and purification was reported previously[31] and the G101V and F104L mutants were previously reported[17]. Briefly, mutants were introduced by PCR using primers with the desired mutation. Bacterial E. coli BL21 cells were transformed with appropriate plasmids and proteins expressed as N-terminal GST fusions by IPTG induction in SuperBroth. Recombinant GST-BCL-2 fusions were purified from cellular proteins by glutathione-agarose resin (Genscript; CAT#L00206) and eluted with 10 mM reduced glutathione. GST-BCL-2 fusions were cleaved overnight at 4 °C with Prescission Protease to remove the GST-fusion and purified to homogeneity by size exclusion chromatography using a Superdex 75 10/300 (GE healthcare) equilibrated in 20 mM Tris pH 8 and 150 mM NaCl.

**Crystallisation**. BCL-2 proteins were incubated with 3 molar excess of the desired compound in DMSO. DMSO was removed by buffer exchange in an Amicon® Ultra-4 10 kDa cut of spin concentrator (Millipore; CAT#UFC801096). Initial crystals were obtained with BCL-2 WT and the desired compound by the hanging drop vapour diffusion method at 291 K with a precipitant solution consisting 5% PEG4K, 40% PEG400, 0.1 M MES pH 6.0. Initial BCL-2 WT: venetoclax analogue crystals were used as crystallisation seeds to streak seed, using a cat's whisker, into a new crystallisation drop containing BCL-2 WT and venetoclax to obtain BCL-2 WT:venetoclax crystals. All subsequent crystallisation experiments with mutants and venetoclax were obtained by streak seeding from BCL-2 WT:venetoclax crystals. Crystallisation conditions were optimised by diluting precipitant concentrations by a factor in the range 0.5–1 with water and varying the pH by 0.2 intervals from the initial condition. BCL-2 G101V:S55746 crystals were obtained by the hanging drop vapour diffusion method with a precipitant solution consisting 1 M NaCl, 0.1 M Succinic acid-NaOH pH 5.0, 10% PEG200. Prior to diffraction experiments crystals were cryoprotected by supplementing the mother liquor with PEG400, glycerol or ethylene glycol, then cooled in liquid nitrogen.

**Structure determination**. All datasets were collected at 100 K at the Australian Synchrotron MX2 beamline using an ADSC quantum 315R CCD detector (BCL-2 WT:venetoclax and BCL-2 F104L:venetoclax only) or an Eiger X 16M direct detector (all other datasets)[32]. Diffraction experiments were processed in XDS[33]

and scaled in either XDS or Aimless[34]. The phase problem was solved by molecular replacement using Phaser[35] and the BCL-2 WT:navitoclax structure chain A with all waters and ligands removed (PDBid 4LVT) as a search model[10]. The model was refined by iterative reciprocal and real space refinement using PHENIX refine[36] and Coot[37], respectively with at least one round of cartesian simulated annealing refinement in PHENIX prior modelling ligands to avoid phase bias. Ligand initial models and restraints for venetoclax and S55746 were generated using the Grade web server (version 1.2.9, Global Phasing Ltd.). For atoms with multiple conformations initial occupancies were set to 0.5 modelling each conformer into appropriate density prior to multiple rounds of refinement in phenix using the refine occupancy function in addition to standard refinement procedures. X-ray data to geometry weights or atomic displacement factors were determined automatically using the optimize X-ray/stereochemistry weight and optimise X-ray/ADP weight respectively. Model validation was performed in Coot and Mol-Probity[38]. Data statistics were calculated in PHENIX using the generate Table 1 for journal function. Stereo images with electron density are displayed in Supplementary Fig. 7.

The BCL-2 G101V:venetoclax data processed and solved in the orthorhombic spacegroup $P2_12_12_1$ with a single protein chain in the assymetric unit. However, during refinement $R_{fact}$ and $R_{free}$ increased in successive refinement cycles, giving final values that were unreasonable when compared to similar resolution structures from the protein data bank. This did not occur when the structure was modelled in the monoclinic spacegroup $P2_1$ with two protein chains in the assymetric unit. As a consequence, the monoclinic model and data were used.

The BCL-2 G101V:S55746 structure processed in the monoclinic spacegroup $P2_1$ with two protein chains in the assymetric unit. The data were anisotropic with data extending to 2.7 Å in the a* and 2.0 Å in the b* and c* directions, and were ellipsoidally truncated and scaled by the diffraction anisotropy server[39] without applying B-factor sharpening. After applying ellipsoidal truncation the data completeness in high resolution shells dropped, with completeness dropping below 90% from 2.5–2.0 Å. Furthermore, the electron density for S55746 in one of the protein chains was more poorly resolved. This copy of S55746 was modelled into the electron density according to the S55746 orientation from the BCL-2 WT:S55746 structure (PDB id 6GL8)[12]. To avoid phase bias no coordinates from the original WT:S55746 structure were used in refinement and orientations were matched by protein alignments and visual inspection only. The analysis presented here relates to the protein chain without this problem.

**Structural analyses**. All crystal structure representations were made using Mac-PyMOL version 1.8.0.3 (Schrödinger LLC). BCL-2 pocket volumes were calculated in a multistep step process. Initially PDB models were stripped of all non-protein atoms and pockets were filled with water molecules using the hollow program (version 1.2)[40] with a 12 Å sphere radius from residue BCL-2 104 and 0.2 grid spacing. BCL-2 pockets were then defined as any water molecule within 5 Å of the venetoclax chlorophenyl (atoms CL and C10) for the P2 pocket or 2 Å from atoms in the venetoclax azaindole (atoms C37–43, N5–6). Water atom selections were visually inspected to remove atoms that may be included in selection criteria but not connected to the desired pocket. The volume of each pocket selection was then determined using the 3vee web server volume assessor function (http://3vee.molmovdb.org /volumeCalc.php)[41] with a 5 Å probe radius and grid resolution set to high.

Alignments to compare structures were performed in MacPyMOL version 1.8.0.3 (Schrödinger LLC) aligning Cα carbons from the α2–5 helices residues

(91–164). In the BCL-2:venetoclax structures there is a crystal contact between R18 and the acyl oxygen from the venetoclax benzamide (a hydrogen bond) in one of the alternate conformations (conformer A). The conformation of the venetoclax conformer A benzamide is equivalent to the conformation of ABT-263 bound to BCL-2 (PDB id 4LVT) which has no crystal contacts to the bound ligand. The published BCL-2 WT:compound 1 structure (PDB id 4MAN) has a crystal contact between the α5–6 loop and the benzine ring from the benzamide moiety of compound 1 from symmetry related molecules that may influence the conformation of compound 1.

Distances between venetoclax cyclohex-1-ene ring atoms (C1–6), ABT-263 atoms (C30, C39, C29, C26, C25 and C28 ordered according to equivalent positions on venetoclax) or compound 1 (C1, C9, C29, C23, C21 and C28) ordered according to equivalent positions on venetoclax) and BCL-2 L137 Cα in Supplementary Table 1 were determined using the distance command in MacPyMOL version 1.8.0.3. For fair comparison the BCL-2 WT:venetoclax cyclohex-1-ene ring A conformer was used as a reference as it matched the conformers from the ABT-263 and G101V:venetoclax structures. The BCL-2 WT: ABT-263 (PDB id 4LVT), BCL-2 WT:compound 1 (PDB id 4MAN) and BCL-2 G101V:venetoclax structures had two copies of protein and drug in the asymmetric unit. Distances for both copies were calculated and mean values presented in the Supplementary Table 1. RMSD values were calculated for each structure relative to the BCL-2 WT:venetoclax cyclohex-1-ene ring A conformer.

**Surface plasmon resonance**. SPR experiments were performed as previously described[17]. Briefly, experiments were performed in HBS-EP buffer consisting 10 mM hepes pH 7.4, 150 mM sodium chloride, 3.4 mM EDTA, 0.005% tween 20 and optionally 1 mM TCEP for BAXBH3 experiments, at 25 °C. Experiments were perfomed on either a BIAcore 4000 or BIAcore S200 using a SA sensor chip (GE healthcare) immobilized with biotinylated BIMBH3 (DMRPEIWIA-QELRRIGDEFNAYYARR) or BAXBH3 (ADASTKKLSECLKRIGDELDSN-MELQRMIAA) peptide, using a BIMBH3–4A peptide (DMRPEIWAAQEARRAGDEANAYYARR) as a non-binding reference. BIMBH3 and BAXBH3 Peptide affinities were determined by direct binding with BCL-2 (0–63 nM) as the analyte. The lowest concentrations used in direct binding experiments were 1 nM for BIAcore 4000 and 0.1 nM for BIAcore S200 instruments, concentrations below these were below the sensitivity thresholds for the instruments. Venetoclax affinity was determined by competition against immobilized BIMBH3 peptide, using BCL-2 (0–250 nM) pre-mixed with venetoclax (0, 20, 40, 60 nM) as the analyte. Direct binding experiments were fitted to a 1:1 binding site kinetic model, fitting on and off rates, in BIAevaluation software. For competition experiments steady-state responses were determined in BIAevaluation software according to the software defaults in affinity evaluation mode averaging responses for 5 sec. Response data were fitted to a steady-state competition model (Equation 1) in Prism 7.0d for mac (GraphPad Software, La Jolla California USA):

$$R = R_{max} \cdot \frac{[BCL2] - \frac{([BCL2]+[VEN]+K_I) \pm \sqrt{([BCL2]+[VEN]+K_I)^2 - (4.[BCL2].[VEN])}}{2}}{K_D + [BCL2] - \frac{([BCL2]+[VEN]+K_I) \pm \sqrt{([BCL2]+[VEN]+K_I)^2 - (4.[BCL2].[VEN])}}{2}} \quad (1)$$

$R$ = response, $R_{max}$ = fixed maximal response, $[BCL2]$ = fixed BCL-2 concentration, $[VEN]$ = fixed venetoclax concentration (or S55746), $K_D$ = fixed BIMBH3 peptide apparent steady-state binding constant, $K_I$ = fitted equilibrium binding constant for venetoclax (or S55746). BIMBH3 $K_D$ and $R_{max}$ for the steady-state competition model were calculated in Prism 7.0d for mac using the data from 0 nM venetoclax from the same experiment, using a one-site specific binding model. All affinity measurements were performed in at least two independent experiments using independent protein preparations.

**Plasmids, retrovirus production and infection**. Wild-type BCL-2 construct was reported previously[31]. Point mutation at BCL-2 Gly101 to Val was introduced using primers (with the desired mutation) using PCR. cDNA encoding either wild-type FLAG-BCL-2 or -BCL-2 mutants were inserted into the MSCV-IRES-hygromycin retroviral construct as previously described[42].

**Cell lines**. KMS-12-PE (sourced from DSMZ in 2013; Cat#ACC606) were cultured with HTRPMI with 10% fetal bovine serum. Early passages (P5-P7) after purchase were cryopreserved and thawed for the experiments. Cells beyond passage 15 were not used. Monthly tests for mycoplasma were consistently negative (MycoAlert mycoplasma detection kit; Lonza, GA, USA). The expression level of WT or G101V BCL2 in KMS-12-PE cells were determined by intracellular FACS using a BCL2 antibody (BCL2–100, WEHI, 1:200)[43] on an LSR-Fortessa flow cytometer.

**Cell viability assays**. To test the sensitivity of the engineered KMS-12-PE inhibitor to S55746, cells were seeded in a 96-well plate at 5000 cells/well and treated with serially diluted concentrations (0–10 μM, 5-point 1:8 dilution) of drug. Cell viability 24 h after treatment was then determined using the CellTiter-Glo assay (Promega, Cat#G9241). Experiments were performed on antibiotic-resistant pools of cells transfected with vectors expressing wild-type, mutant BCL-2 or the empty vector control. Experiments were performed on three separate occasions; the data is shown as the means ± SD of those three independent experiments.

**Reporting summary**. Further information on research design is available in the Nature Research Reporting Summary linked to this article.

## Data availability

The refined coordinates and data for all structures were deposited in the PDB: BCL-2: venetoclax PDB id 6O0K, BCL-2 G101V:venetoclax PDB id 6O0L, BCL-2 G101A: venetoclax PDB id 6O0P, BCL-2 F104L:venetoclax PDB id 6O0M and BCL-2 G101V: S55746 PDB id 6O0O. The source data underlying Fig. 3a–h, Fig. 4a and Table 2 are provided as a Source Data File. A reporting summary for this Article is available as a Supplementary Information file. All other data supporting the findings of this study are available from the corresponding author on reasonable request.

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

## Acknowledgements

Research in the authors' laboratory is supported by NHMRC project grants (1059331 and 1079706), program grants (11131233 DCSH, PMC; 1113577 AWR) and fellowships (1156024 DCSH, 1079560 AWR, 1116934 PMC, 1079700 PEC), the Leukemia and Lymphoma Society (SCOR 7015-18 and Fellowship 5467-18 to RT), the Victorian Cancer Agency (Fellowship to IJM), project support from The Cancer Council of Victoria (Project 1124178 to IJM), the Victorian State Government Operational Infrastructure Support, and the Australian Government NHMRC IRIISS. This research was undertaken in part using the MX2 beamline at the Australian Synchrotron, part of ANSTO, and made use of the Australian Cancer Research Foundation (ACRF) detector. The authors would like to thank Yan-Hong Tan for technical assistance, Dr W. Douglas Fairlie and Dr Erina Lee for providing BCL-2 plasmid DNA and the Collaborative Crystallisation Centre (C3, CSIRO) for crystallisation screening. We would also like to thank Wayne Fairbrother for discussions related to the manuscript. Finally the authors would like to thank the following cats and their owners for donating their naturally shedded whiskers for crystallisation seeding experiments: Tilly, Boris, Patsy and Snoop Catty Cat.

## Author contributions

R.W.B., M.A.A., P.B., I.J.M., A.W.R., D.C.S.H., P.M.C. and P.E.C. made substantial contributions to the conception or design of the work; R.W.B., J.-N.G., C.S.L., D.L., C.A.W., R.T., P.E.C. to the acquisition of data; R.W.B., J.-N.G., P.M.C., P.E.C to analysis of data; R.W.B., G.L., A.W.R., D.C.S.H., P.M.C., P.E.C. to the interpretation of data; R.W.B., A.W.R., D.C.S.H., P.M.C. and P.E.C. drafted the work or substantively revised it.
