## [Peer Review File · Nature Communications]

Reviewers' comments:

Reviewer #1 (Remarks to the Author):

The manuscript by Birkinshaw et al. describes the structure of BCL-2 in complex with venetoclax and how point mutations developed by chronic lymphocytic leukemia patients treated with the drug enable resistance to venetoclax. This work builds on the recent article by Blombery et al. which first described the G101V mutation. BCL-2 is a well-known apoptosis regulator whose overexpression is commonly associated with tumor maintenance, progression, and chemoresistance. As such, inhibition of BCL-2 is of paramount importance in the treatment of several cancers, and multiple inhibitors have been developed, though to my knowledge venetoclax is the only inhibitor currently FDA-approved. The authors provide an in-depth comparison of the venetoclax binding mode with other inhibitors, in complex with WT BCL-2 and BCL-2 mutants. Based on this structural data, they singled out the repositioning of E152, due to the bulk introduced by the glycine to valine mutation, as the main culprit for loss of affinity of BCL-2 to venetoclax. They confirmed this hypothesis by testing several mutations (G101A, E152A, F104L, F104C) and compared binding of other inhibitors to these mutants as well. These data improve our understanding of BCL-2 inhibition and escape mechanisms. The manuscript should enable structure-based development of BCL-2 inhibitors which can also bind to the G101V mutant.

While the work is of great interest for the cancer research community, some improvements could be made to the text and figures, in particular concerning the description of data refinement and analysis.

- 1) Even though there is mention on Fig. 2 of referring to methods for details, there are no explanations given concerning the pocket volume calculations used in the text. These calculations are not straightforward and should either be described in the methods, refer to a manuscript describing the method used, or if using a home script it could be added as a supplemental file and/or as a link to an archive server.
- 2) Many multiconformers are present in the structures used, and are highly relevant to the conclusions drawn. Refinement of these multiconformers should be described in the methods (occupancy starting point, options used for refinement – in particular weight optimization for geometry / data, etc).
- 3) Several structures are compared while belonging to different space groups & unit cells. Please comment on the possible influence of crystal packing on the binding site.

4) The structure of S55746 in complex with WT BCL-2 (pdb 6GL8) was solved at 1.4Å resolution. In this structure, the electron density for S55746 is very well-defined and clearly supports the model. Based on Suppl. Fig. 1h., the electron density for S55746 in complex with BCL-2 G101V isn't so well-defined. I am assuming that the electron density (ED) in Suppl. Fig. 1h is that of the better-defined ED, as the second chain issue is addressed in the methods. Based on that ED, Fig. 4b could be misleading. How was S55746 modelled and refined? Was pdb 4LVT used for phasing and S55746 modelled later on? If the structure was solved using 6GL8 as molecular replacement model and barely refined after that, the position of S55746 would be highly biased to support the authors' claim. Please confirm that efforts have been made not to bias the structure, e.g. starting refinement with a different positioning of S55746 resulting in poorer structure quality indicators or S55746 "moving back" to the current position.

5) The validation reports show that Y18 and E152 are clashing in several structures (in particular WT BCL-2 in complex with Venetoclax) and / or issues with the geometry of Y18. Since these are high resolution structures and much of the interpretation relies on the position of these residues, please improve the model or provide an explanation for these issues. Furthermore, many clashes in the validation report are coming from water molecules which are likely being overmodelled, and I would remove those unless the electron density strongly suggests otherwise.

6) On Fig.1. the labeling of the α helices is confusing as they are barely visible and not described in the legend. I suggest removing the helix labeling and keeping the BCL-2 structure opaque, and possibly adding an extra panel showing the structure in absence of ligand with more transparency of the surface and helix labels.

7) Many figures (e.g. all panels on Fig. 2 and Suppl. Fig. 3, as well as Fig. 4b&d) could be improved by using the "set cartoon_side_chain_helper" option in Pymol as it would remove some visual clutter and focus the reader's attention on the relevant points. In absence of side chain, e.g. for G101, the residue of interest could be pointed out by coloring it in a similar but slightly different color (for example a lighter or darker green on Suppl. Fig. 3).

8) Fig. 5 would be more relevant if Bax-BH3 was binding to BCL-2 G101V. Since based on table 2 the affinity of BCL-2 G101V to Bax-BH3, though decreased compared to WT, still is in the low nM range, were attempts at crystallizing the BCL-2 G101V / Bax-BH3 complex made? This structure would be informative especially as the authors write "BCL-2 G101V maintains affinity for the BH3 motif of pro-apoptotic proteins such as BAX and BIM, and thus can still function to suppress apoptosis" even though the affinity is reduced by ~10 fold and G101 is part of the conserved BH3 motif.

9) In Suppl. Fig. 2, please indicate the distance range (5-6Å) for the Venetoclax ring to L137 in BCL-2 WT. A 1Å movement is more significant if the distance between residues is small.

10) Suppl. Fig. 4: In some instances it seems that the 1:1 binding model does not fit well, especially at the highest concentration. How do the affinity and quality of fit change if this highest concentration is removed? What concentrations have been used (a colored legend would be useful)? Please add a supplementary table with the calculated K_{on} and K_{off} .

11) Suppl. Fig. 5: please present the data for B in the same way as for the other constructs. Please indicate how many data points or how much time was used for the averaging of the SPR response.

I would also like to mention some minor typos and missing abbreviations I have noticed while reading:

- In Fig.4a, the EV abbreviation is not explained. G101V is written as G010V in the legend.
- The CLL abbreviation is not introduced.
- Bona fide is written as bone fide in the introduction.
- Please change “more mild” to “milder”.
- A “of” is missing after the K_i of BCL-2 WT to S55746.
- In the crystallization part of the methods, “varying” is spelled “vaying”.
- In the structure determination part of the methods, the phase problem was solved “by” molecular replacement; “monoclinic” is spelled several times as “monoclinc” and “furthermore” is spelled “futhermore”.
- Please replace “BCL2” by “BCL-2” in the figures and Table 2 for consistency.

Nadia Leloup

Reviewer #2 (Remarks to the Author):

In this study Birkinshaw et al provide novel structural insight on the interaction of BCL-2 with venetoclax. Thereby, they build on the observation that the BCL2-G101V mutation is associated with resistance to venetoclax, as they described previously (Blombery et al, Ref 17).

By solving the crystal structure of BCL-2 in complex with venetoclax and comparing this to previously published structures of BCL-2 complexed with analogues of venetoclax, they find subtle but important differences in the conformation of the binding site. Further, they solved the crystal structures of the BCL-2 F104L, G101V and G101A mutants and compared the binding of venetoclax. Of note, they also included a competitor compound, S55746, and compared the binding affinities and structural conformations, revealing important differences in the binding moieties of both compounds.

This is a convincing paper that will be highly relevant to the field, in particular in regards to the structural understanding of acquired resistance to venetoclax as observed in CLL patients upon prolonged treatment.

Specific comments:

In Figure 1b the authors describe two distinct conformations of the venetoclax BA and 4DM moieties binding to BCL-2 (flipping of the 4DM ring). I believe that this is visualized by arrows, but this may be more easy to understand if these alternate conformations were shown side-by-side.

The figure legends (in particular Figure 2) are sometimes difficult to follow and could be improved by providing more details e.g. in line 471 "b-d Venetoclax": should this not read b-f? Maybe use different symbols (e.g. arrowheads) to distinguish between key residues or movements.

Why is G101A shown in Figure 3(e) and not G101V with Venetoclax? Is this the same set of data previously described in reference 17?

Meike Vogler

Reviewer #3 (Remarks to the Author):

NCOMMS-19-06466-T article by Richard Birkinshaw, Peter Czabotar and colleagues is a timely structure-based study that reveals mechanistically how drug-resistant mutants in BCL-2 impede binding of several BCL-2-targeting drugs of interest, including the FDA-approved drug, venetoclax. Among 5 new structures the authors present the first crystal structures of venetoclax with WT and mutant BCL-2. The manuscript is well written, easy and fun to read, and contains a wealth of information of interest to the apoptosis field, drug designers, and clinicians, and therefore should be published without delay. For instance, the authors predict engineering of venetoclax at the P2 pocket possibly allowing better binding to drug-resistant mutant G101V BCL-2.

I have on minor comment and several edits.

1. Related to Figure 4. To be thorough the authors should include a western blot showing that WT and G101 BCL-2 proteins are expressed at the same level in KMS-12-PE cells, as claimed.

Edits:

Line 300 The phase problem was solved by molecular...

Line 458 Chemical structure of venetoclax (Ven) – keep venetoclax throughout rather than capitalizing first letter

Line 460 (4DM), the 5-5-dimethylcyclohex-1-ene

Line 481 Data are representative

Line 484 WT and G101 mutant BCL-2 proteins were expressed at similar levels. In vitro sensitivity to S55746 (0-10 uM) was measured...

Line 506 Mean values are from at least

Line 510 BCL-2 mutants with venetoclax

Review response

We thank the reviewers for their positive appraisal of our paper and its relevance. Below we detail our responses to their queries and comments, which, now addressed, have improved the manuscript.

Reviewer #1

1) Even though there is mention on Fig. 2 of referring to methods for details, there are no explanations given concerning the pocket volume calculations used in the text.

We thank the reviewer for spotting this accidental omission from our methods. We have now included a comprehensive structural analyses methods section detailing how the pocket volumes were calculated. Additionally, we have included a detailed description for how distances in supp. Fig 2 and supp. Table 1 were calculated. See lines (332-358).

2) Many multiconformers are present in the structures used, and are highly relevant to the conclusions drawn. Refinement of these multiconformers should be described in the methods.

We have now included statements in the methods to clarify these points see lines 307-313.

3) Several structures are compared while belonging to different space groups & unit cells. Please comment on the possible influence of crystal packing on the binding site.

All the venetoclax structures crystallised in similar conditions and the same space group, with the same crystal contacts. As we discuss in the methods (lines 313-317) the BCL-2 G101V:venetoclax was modelled in the lower symmetry spacegroup $P 2_1$, due to improved refinement statistics.

The S55746 structure with G101V was in a different spacegroup to the previously published 6GL8 structure. We do not claim differences in binding mode between these structures.

Comparison of the BCL-2 WT:venetoclax, BCL-2 WT:ABT-263 and BCL-2 WT:compound **1** structures compares 3 different spacegroups. There are no crystal contacts to ABT-263, however there are contacts crystal contacts to compound **1** and venetoclax in the benzamide region. We do not discuss differences between these structures in detail in the benzamide region.

We have now inserted comments on this in the structural analyses methods section to clarify (lines 341-348). These comments do not change our interpretation or analysis of the structures.

4) The structure of S55746 in complex with WT BCL-2 (pdb 6GL8) was solved at 1.4Å resolution. In this structure, the electron density for S55746 is very well-defined and clearly supports the model. Based on Suppl. Fig. 1h., the electron density for S55746 in complex with BCL-2 G101V isn't so well-defined. I am assuming that the electron density (ED) in Suppl. Fig. 1h is that of the better-defined ED, as the second chain issue is addressed in the methods. Based on that ED, Fig. 4b could be misleading. How was S55746 modelled and refined? Was pdb 4LVT used for phasing and S55746 modelled later on? If the structure was solved using 6GL8 as molecular replacement model and barely refined after that, the position of S55746 would be highly biased to support the authors' claim. Please confirm that

efforts have been made not to bias the structure, e.g. starting refinement with a different positioning of S55746 resulting in poorer structure quality indicators or S55746 “moving back” to the current position.

4LVT chain A was used for molecular replacement, for all molecular replacement solutions all waters and ligands were removed prior to searching, to remove potential phase bias. The S55746 model and restraints were generated by the grade server and modelled into the unbiased density after cartesian simulated annealing refinement of the refined search model from phaser.

To clarify S55746 was modelled in the 6GL8 conformations by alignment and visual inspection only. No coordinates from the 6GL8 model were directly used in molecular replacement or refinement at any stage and so phase bias towards the 6GL8 structure is not possible.

We have amended our methods to clarify see lines 305, 307-309, 327-329.

5) The validation reports show that Y18 and E152 are clashing in several structures (in particular WT BCL-2 in complex with Venetoclax) and / or issues with the geometry of Y18. Furthermore, many clashes in the validation report are coming from water molecules which are likely being overmodelled, and I would remove those unless the electron density strongly suggests otherwise.

We have addressed these points from the BCL-2 WT:venetoclax PDB validation report, we have resubmitted coordinates to the PDB and have attached the updated validation report. The changes do not significantly affect interpretation of the models. We have updated figures describing the positioning of Y18 in the WT structure to accommodate and subtle changes that may have occurred in addressing this issue. We have rechecked electron density for the highlighted waters and have removed those for which electron density is borderline. We have also updated statistics in Table 1 to reflect the re-refined model.

6) On Fig.1. the labeling of the α helixes is confusing as they are barely visible and not described in the legend. I suggest removing the helix labeling and keeping the BCL-2 structure opaque, and possibly adding an extra panel showing the structure in absence of ligand with more transparency of the surface and helix labels.

We have removed the helix labelling to make figure 2 clearer. We prefer to not add an additional panel as this will make the figure too cluttered so have focused on the surface as this is more relevant to our discussion of the structures (which does not reference helix numbering). Helix numbering is still present on Figure 2 as appropriate for discussion in the text.

7) Many figures (e.g. all panels on Fig. 2 and Suppl. Fig. 3, as well as Fig. 4b&d) could be improved by using the “set cartoon_side_chain_helper” option in Pymol as it would remove some visual clutter and focus the reader’s attention on the relevant points....

We have removed stick representation from backbone residues where appropriate in the suggested figures. We felt these changes were subtle did not achieve the desired outcomes so have removed some cartoon representations to highlight the G101V mutation.

8) *Fig. 5 would be more relevant if Bax-BH3 was binding to BCL-2 G101V. Since based on table 2 the affinity of BCL-2 G101V to Bax-BH3, though decreased compared to WT, still is in the low nM range, were attempts at crystallizing the BCL-2 G101V / Bax-BH3 complex made? ...*

We have tried to obtain crystals for BCL-2 mutants and Bax-BH3. Unfortunately, crystallising BCL-2 is challenging and although we have been fortunate to describe many structures in this paper it is not always possible to obtain crystals. Despite several years of trying we have been unsuccessful in replicating the crystals originally described by Ku et al. with BCL2 WT bound to Bax-BH3 (PDB id 2XA0) and have not been able to crystallise BCL-2 G101V with Bax-BH3 either. We agree that this structure would be interesting but have thus far been unable to obtain it.

9) *In Suppl. Fig. 2, please indicate the distance range (5-6Å) for the Venetoclax ring to L137 in BCL-2 WT. A 1Å movement is more significant if the distance between residues is small.*

These values are in supplemental table 1, but we have indicated the range on the figure as suggested and the figure legend amended accordingly (lines 580-582)

10) *Suppl. Fig. 4: In some instances it seems that the 1:1 binding model does not fit well, especially at the highest concentration. How do the affinity and quality of fit change if this highest concentration is removed? What concentrations have been used (a colored legend would be useful)? Please add a supplementary table with the calculated K_{on} and K_{off} .*

We have included supplemental table 2 detailing parameters for the fits from the 3 independent experiments that were used to derive the K_D values for peptide binding in table 2 (lines 612 to 616). Although protein concentrations were indicated in the methods section we have now included in supp Fig. 4 and the legend has been amended to include this addition (lines 592-593).

Respectfully we disagree with the reviewer here on removal of data. The fits shown are appropriate as reflected in χ^2 values $>10\%$ (mostly $>5\%$) of R_{max} . There are some minor deviations late in the association phase at higher concentration, however the dissociation phases fit very well for these higher concentrations. It is beneficial to include these data for accurate determination of dissociation rates as larger changes in dissociation response occur at higher concentration, which improves accuracy of fitting the dissociation phase.

Furthermore, it is clear from the now included supp. Table 2 that poorer fitting (higher χ^2 to R_{max} ratios) has little impact on binding affinity between independent experiments and is appropriate for our interpretation of differences between mutants.

11) *Suppl. Fig. 5: please present the data for B in the same way as for the other constructs. Please indicate how many data points or how much time was used for the averaging of the SPR response.*

We have updated the methods to include the averaging time of 1 sec as per the defaults in the BIAevaluation software (lines: 369-371) and have updated supp. Fig 5. As requested.

I would also like to mention some minor typos and missing abbreviations I have noticed while reading:

- *In Fig. 4a, the EV abbreviation is not explained. G101V is written as G010V in the legend.*

- *The CLL abbreviation is not introduced.*
- *Bona fide is written as bone fide in the introduction.*
- *Please change “more mild” to “milder”.*
- *A “of” is missing after the Ki of BCL-2 WT to S55746.*
- *In the crystallization part of the methods, “varying” is spelled “vaying”.*
- *In the structure determination part of the methods, the phase problem was solved “by” molecular replacement; “monoclinic” is spelled several times as “monoclinc” and “furthermore” is spelled “futhermore”.*
- *Please replace “BCL2” by “BCL-2” in the figures and Table 2 for consistency.*

We have made these suggested changes.

Reviewer #2 (Remarks to the Author):

Specific comments:

1) In Figure 1b the authors describe two distinct conformations of the venetoclax BA and 4DM moieties binding to BCL-2 (flipping of the 4DM ring). I believe that this is visualized by arrows, but this may be more easy to understand if these alternate conformations were shown side-by-side.

We have amended figure 1b as suggested to show the two distinct venetoclax conformers and amended the figure legend appropriately (lines 518 and 520-521).

2) The figure legends (in particular Figure 2) are sometimes difficult to follow and could be improved by providing more details e.g. in line 471 “b-d Venetoclax”: should this not read b-f? Maybe use different symbols (e.g. arrowheads) to distinguish between key residues or movements.

We have revised figure 2 using dashed arrows for movements and solid arrows for key residues. We have also amended the Figure legend to improve our description of the figure (lines:527-531).

3) Why is G101A shown in Figure 3(e) and not G101V with Venetoclax? Is this the same set of data previously described in reference 17?

The G101A binding data is new to this paper, whilst G101V binding data has previously been described in reference 17. We do not describe the G101V binding data as a consequence and indicate this in table 2. We could include a representative experiment from an independent experiment for the G101V venetoclax data that has not previously been published, however, we feel this would be misleading to the reader.

Reviewer #3 (Remarks to the Author):

1. Related to Figure 4. To be thorough the authors should include a western blot showing that WT and G101 BCL-2 proteins are expressed at the same level in KMS-12-PE cells, as claimed.

We have included FACS profiles in figure 4 to show that proteins are expressed at similar levels and have amended the cell lines methods section to account for the additional experiment lines 388-390.

Edits:

Line 300 The phase problem was solved by molecular...

Line 458 Chemical structure of venetoclax (Ven) – keep venetoclax throughout rather than capitalizing first letter

Line 460 (4DM), the 5-5-dimethylcyclohex-1-ene

Line 481 Data are representative

Line 484 WT and G101 mutant BCL-2 proteins were expressed at similar levels. In vitro sensitivity to S55746 (0-10 uM) was measured...

Line 506 Mean values are from at least

Line 510 BCL-2 mutants with venetoclax

We have made these changes.

REVIEWERS' COMMENTS:

Reviewer #1 (Remarks to the Author):

Many thanks to the authors for their detailed replies and adjustments to text and figures.

I have two more remarks building up on the authors' response & adjustments:

1) The authors' reply to question 8: "We have tried to obtain crystals for BCL-2 mutants and Bax-BH3. Unfortunately, crystallising BCL-2 is challenging and although we have been fortunate to describe many structures in this paper it is not always possible to obtain crystals. Despite several years of trying we have been unsuccessful in replicating the crystals originally described by Ku et al. with BCL2 WT bound to Bax-BH3 (PDB id 2XA0) and have not been able to crystallise BCL-2 G101V with Bax-BH3 either. We agree that this structure would be interesting but have thus far been unable to obtain it." is noteworthy of being included in a modified form (including why the structure would be useful) somewhere in the text, ideally shortly in the discussion.

2) I agree with the authors based on the novel suppl. Table 2 that the poorer fitting (higher χ^2 to Rmax ratios) seems to have little impact on binding affinity between independent experiments in this case. However, I am not entirely satisfied with the quality of the SPR data. I wonder why the lowest concentration measured is 1nM when the KDs are all subnanomolar, and on suppl. Fig. 4a. (binding of BCL-2 G101A to BIMBH3) the 1nM concentration does not seem to show binding (or is the figure unjustly unclear?) when this concentration should be >5x reported KD (see for comparison Fig. 4b. of BCL-2 F104C binding to BIMBH3 shows clear binding at 1nM with a reported KD in the same range as BCL-2 G101A). I believe a longer association time would have been beneficial to the quality of data for the BCL-2 G101A / BIMBH3 interaction, as it seems only the linear part of the response has been measured. Please comment on these possible limitations of the SPR analysis & address them in the text if necessary.

I am otherwise happy to recommend publication of this work.

Nadia Leloup

Reviewer 1 response

Below is our response to reviewer 1's specific comments:

1) **The authors' reply to question 8:** *"We have tried to obtain crystals for BCL-2 mutants and Bax-BH3. Unfortunately, crystallising BCL-2 is challenging and although we have been fortunate to describe many structures in this paper it is not always possible to obtain crystals. Despite several years of trying we have been unsuccessful in replicating the crystals originally described by Ku et al. with BCL2 WT bound to Bax-BH3 (PDB id 2XA0) and have not been able to crystallise BCL-2 G101V with Bax-BH3 either. We agree that this structure would be interesting but have thus far been unable to obtain it."* **is noteworthy of being included in a modified form (including why the structure would be useful) somewhere in the text, ideally shortly in the discussion.**

We have now included an additional sentence in lines 218 and 219 to address this request.

Our initial manuscript did not discuss the G101V mutation in the context of BAX discussion in detail as we did not have the structure to analyse. We feel including more detail than proposed would be misleading to the reader.

2) I agree with the authors based on the novel suppl. Table 2 that the poorer fitting (higher χ^2 to Rmax ratios) seems to have little impact on binding affinity between independent experiments in this case. However, I am not entirely satisfied with the quality of the SPR data.

We are pleased the reviewer now agrees with us on the appropriateness of our fitting. However, we feel the reviewer is being overly critical in stating "I am not entirely satisfied with the quality of the SPR data." The reviewer is commenting on the BCL-2 direct binding to BH3 peptides in suppl fig 4. The conclusion from these experiments is that BCL-2 WT and the mutants tested maintain high affinity binding to BIMBH3 and BAXBH3 peptides. We have previously published this is the case for the G101V and F104L mutations (Blomberg et al 2018) and the data in suppl fig 4 support the same conclusions for the G101A, F104C, E152A and G101V E152A double mutant reported in our study. We would note that we have been judicious in our discussion of all our data relative to experimental limitations.

2) cont.

I wonder why the lowest concentration measured is 1nM when the KDs are all subnanomolar, and on suppl. Fig. 4a. (binding of BCL-2 G101A to BIMBH3) the 1nM concentration does not seem to show binding (or is the figure unjustly unclear?) when this concentration should be >5x reported KD (see for comparison Fig. 4b. of BCL-2 F104C binding to BIMBH3 shows clear binding at 1nM with a reported KD in the same range as BCL-2 G101A).

Most of our experiments were performed on a BIAcore 4000 instrument this instrument did not have sufficient sensitivity to accurately fit BCL-2 concentrations below 1 nM.

The reviewer has noticed that the experiment in question, G101A BIMBH3 binding, was performed on a different instrument (BIAcore S200) than the other experiments (BIAcore 4000). These machines have slightly different surface properties and instrument sensitivities, notably the S200 is more sensitive than the 4000. In the G101A BIMBH3 interaction in

question, using the S200 allowed us to use a lower concentration range 0.1 – 25 nM with 9 curves fitted. We inadvertently mislabelled these the same as the BIAcore 4000 experiments (1-63 nM, 7 curves fitted). We thank the reviewer for raising our attention to this and have now corrected the labelling and the figure legend.

In relation to the reviewers concern, the now correctly labelled lower concentrations and increased number of curves are indeed more appropriate for the fitting and K_D values for this experiment. However, the change in x-axis scale and response sensitivity means it is challenging to see the curvature in the BCL-2 G101 0.1 nM sensorgram, as the concentration, as compared to the F104C which the reviewer compares this to, is 10-fold lower.

We have amended the supp fig 4 legend to indicate the difference in instruments. We have also amended supp fig 4a to include the actual concentration range for panel a.

2) cont.

I believe a longer association time would have been beneficial to the quality of data for the BCL-2 G101A / BIMBH3 interaction, as it seems only the linear part of the response has been measured.

The third independent experiment for G101A in supp table 2 was performed on the BIAcore 4000, as is the case for other mutants. Although there are slight (2-fold) differences in the on and off rates, the K_D values are comparable to the experiments performed on the BIAcore S200 (experiments 1 and 2). In this instance fitting to the linear part of the response gives equivalent K_D s from experiments performed on different machines. Here is a representative image for the experiment performed on the BIAcore 4000 experiment to confirm:

G101A binding to BIMBH3 on BIAcore 4000 from independent experiment 3 in supp table 2.

We do not feel this needs to be included in the manuscript as the data included in supp fig 4a is sufficient for our conclusions, as we have discussed here.

“Please comment on these possible limitations of the SPR analysis & address them in the text if necessary.”

We have amended the methods section to indicate loss of sensitivity below 1 nM concentrations of BCL-2 on a BIAcore 4000 instrument (lines 377-379).

We have also clarified in the supp fig 4 legend that panel a was performed on a BIAcore S200 and all other panels on a BIAcore 4000.

We have also amended supp fig 4a to have the actual concentration range of 0.1 – 25 nM.